# Syndromic Testing—The Evaluation of Four Novel Multiplex Real-Time Polymerase Chain Reaction Panels

**DOI:** 10.3390/diagnostics15101228

**Published:** 2025-05-13

**Authors:** Mesut Yılmaz, Selcuk Kilic, Fatma Bayrakdar, Selin Nar Ötgün, Ayse Istanbullu Tosun, Umit Zeybek, Faruk Çelik, Gokhan Aygun, Birol Safak, Naim Mahroum

**Affiliations:** 1Infectious Diseases and Clinical Microbiology, International School of Medicine, Istanbul Medipol University, Istanbul 34810, Turkey; myilmaz@medipol.edu.tr; 2Public Health General Directorate, Department of Microbiology Reference Laboratories and Biological Products, Ankara 06430, Turkey; selcuk.kilic@sbu.edu.tr (S.K.); fbayrakdar@windowslive.com (F.B.); selin.narotgun@saglik.gov.tr (S.N.Ö.); 3Department of Clinical Microbiology, School of Medicine, Istanbul Medipol University, Istanbul 34810, Turkey; atosun@medipol.edu.tr; 4Aziz Sancar Institute of Experimental Medicine, Istanbul University, Istanbul 34093, Turkey; uzeybek@istanbul.edu.tr (U.Z.); celikfaruk@yahoo.com (F.Ç.); 5Infectious Diseases and Clinical Microbiology, Cerrahpasa Medical Faculty, Istanbul University, Istanbul 34810, Turkey; gokhan.aygun@iuc.edu.tr; 6Department of Clinical Microbiology, Faculty of Medicine, Namik Kemal University, Tekirdag 59030, Turkey; bsafak@nku.edu.tr

**Keywords:** syndromic testing, real-time PCR, syndromic multiplex panels

## Abstract

**Background:** If used in the right clinical context, PCRs carry great potential in rapidly diagnosing various infectious diseases. **Objectives:** We aimed to evaluate the clinical performance of four novel multiplex real-time PCR (qPCR) assays in the direct detection of pathogens in whole blood, cerebrospinal fluid, respiratory specimens, and stool samples. **Methods:** Spiked negative clinical specimens were used for the evaluation. Clinical samples for the comparative assessment of culture and molecular analyses were simultaneously examined. RINA^TM^ robotic nucleic acid isolation system and Bio-Speedy^®^ multiplex qPCR panels (Bioeksen R&D Technologies, Turkey), and the LightCycler^®^ 96 Instrument (Roche, USA) were used for the molecular testing. **Results:** No qPCR assays produced positive results for the samples spiked with the potential cross-reacting pathogens. The limit of detection (LOD) of the assays changed with the use of 10 and 100 pathogens/mL sample based on the target and sample type. The relative sensitivity and specificity of the assays were, respectively, 82% and 94% for blood, 97.1% and 99.3% for blood culture, 94% and 98% for stool, 96% and 97% for CSF, and 97% and 96% for respiratory specimens. **Conclusions:** The panels evaluated allow the direct molecular analysis of 10 samples from four clinical syndromes on the same run in 3 h with high clinical performance. The number and variety of samples in a single run enable healthcare providers to rapidly and efficiently diagnose and treat various infections.

## 1. Introduction

Since their advent more than a decade ago, the commercial panel-based molecular diagnostics for rapid pathogen detection in different biological samples have transformed clinical microbiology and practice [1]. Classified under syndromic panel testing, these molecular assays simultaneously detect and identify multiple pathogens associated with various syndromes related to bloodstream, respiratory, gastrointestinal (GI), or central nervous system (CNS) infections, which save valuable time and potentially improve healthcare outcomes. In practical life, for patients presenting with suspected infectious disease, with findings that overlap among numerous infectious agents (bacteria, viruses, and other pathogens), syndromic testing provides a simultaneous testing of a high number of pathogens from different biological samples [2]. The results detected aid in the rapid application of directed treatment, halt the misuse of antimicrobials or provide a much wider spectrum of coverage, which eventually enhances stewardship and decreases the risk of resistance among pathogens. What leaves no doubt is that this method of testing saves time, and resources, and enhances the efficacy of microbiological lab processes when compared to old and standard methods.

Nevertheless, molecular syndromic testing technologies also present several challenges, including cost, strategies of use, and the interpretation of results. For instance, current clinical practice guidelines do not always provide clear directions for result interpretation, and clinicians may not be familiar with all the detected organisms and/or resistance genes [3]. Additionally, fixed panel composition can be challenging in terms of application in clinical practice. The closed-system multiplex platforms also carry a contamination risk that may be difficult to identify. Additional challenges include integrating multiplex panels into laboratory workflows and monitoring result accuracy post-implementation.

As the use and application of syndromic testing are expected to continue to become more common, understanding the performance characteristics and limitations of multiplex assays is crucial. Therefore, we aimed to analyze the clinical performance of four novel multiplex real-time PCR (qPCR) assays for the identification of pathogens in clinical practice.

## 2. Materials and Methods

The study was conducted as a prospective observational diagnostic accuracy study. Samples were collected prospectively from patients with suspected infections, and each sample was tested both by culture-based gold standard methods and the novel qPCR panels. Trained clinical personnel at the participating centers collected the samples. For all molecular analyses, processing began within 2–3 h after collection. The detailed process is presented in the following manner:

### 2.1. Molecular Syndromic Testing

The RINATM-M14 robotic nucleic acid isolation system, Bio-Speedy^®^ multiplex qPCR panels (Bioeksen, Sarıyer, Turkey), and LightCycler^®^ 96 Instrument (Roche, Indianapolis, IN, USA) were used for all molecular syndromic tests in our study. Samples of blood, CSF, nasopharyngeal wash/aspirate, sputum, and bronchoalveolar lavage (BAL) were directly loaded into RINATM-M14 nucleic acid extraction cartridges. Oropharyngeal and nasopharyngeal swabs and approximately 30 mg of stool samples were transferred into 500 µL molecular grade water and homogenized before loading. The 75 min extraction protocol was employed for all extractions in the RINATM-M14 robot. Deionized water served as the negative control in each run.

The multiple targets in the pre-loaded and ready-to-use 8-well qPCR strips of the Bio-Speedy^®^ qPCR panels are listed in Table 1, Table 2, Table 3, Table 4 and Table 5. A reaction containing a human DNA-targeted oligonucleotide set was used as an internal control to assess DNA extraction and PCR inhibition [4]. For each qPCR well, 5 µL of the nucleic acid extract was loaded into a qPCR well containing 15 µL of the target-specific multiplex qPCR mixture. The 90 min qPCR protocol was used for all assay types.

### 2.2. Reference Sample Preparation

Reference strains and clinical isolates from culture-confirmed cases were obtained from various collection sites: The Department of Microbiology Reference Laboratories and Biological Products of Public Health General Directorate (HSGM), Istanbul Medipol University (IMU), the Aziz Sancar Institute of Experimental Medicine Istanbul University (DETAE), Istanbul University-Cerrahpasa Medical Faculty (CTF), and Namik Kemal University (NKU). Additional reference strains/materials were purchased from the American Type Culture Collection (ATCC) or NIBSC.

Reference strains: *Corynebacterium diphtheriae*, *Neisseria pharynges*, *Moraxella catarrhalis*, *Streptococcus mutans*, *Streptococcus agalactiae*, *Streptococcus pyogenes*, *Staphylococcus haemolyticus*, *Staphylococcus epidermidis*, *Staphylococcus hominis*, *Staphylococcus simulans*, *Staphylococcus capitis*, *Staphylococcus lugdunensis*, *Stenotrophomonas maltophilia*, *Brucella melitensis*, *Brucella abortus*, *Brucella suis*, *Cryptosporidium parvum*, *Cryptosporidium hominis*, *Salmonella enterica* serovar *Enteritidis*, *S. enterica serovar Typhimurium*, *S. enterica serovar Infantis*, *Campylobacter jejuni*, *Campylobacter coli*, *Aspergillus fumigatus*, *Aspergillus flavus*, *Aspergillus niger*, *Aspergillus terreus*, *Fusarium solani*, *Fusarium oxysporum*, *Fusarium fujikuroi*, *Trichosporon asahii*, *Trichosporon asteroides*, *Trichosporon cutaneum*, *Trichosporon inkin*, *Trichosporon mucoides*, *Trichosporon ovoides*, *Enterococcus faecalis*, *Enterococcus faecium*, *Pseudomonas maltophilia*, *Serratia marcescens*, *Proteus mirabilis*, and *Proteus vulgaris*.

The vector DNAs carrying the target DNA fragments were used as qPCR quantification standards [5]. The vectors were synthesized by GenScript (860 Centennial Ave, Piscataway, NJ 08854, USA). The vector quantity was checked using a 2100 Bioanalyzer (5301 Stevens Creek Blvd. Santa Clara, CA 95051, USA). Standard curves were generated using qPCRs that contained vector DNA copies between 10^6^ and 10^0^, with quantification cycle (Cq) values between 10 and 40. The standard dilutions and the extracted nucleic acid samples were run in duplicate for the quantification.

For all the target and non-target strains, 10^0^–10^5^ genome/mL dilutions in phosphate-buffered saline (PBS) were prepared. Clinical samples confirmed as negative via culture/PCR were spiked with the dilutions in PBS to obtain reference samples. The concentrations were confirmed via qPCR quantification.

### 2.3. Analytical Sensitivity and Specificity as Well as Repeatability

To determine the limit of detection (LOD) in PBS, 10^0^–10^4^ genome/mL dilutions were tested 12 times in the same run. Clinical samples containing 0, 0.1xLOD, 0.5xLOD, LOD, 10xLOD were prepared. In order to evaluate repeatability (precision), each dilution was tested two times in the same run; two different operators performed same-day testing; the tests were repeated on three consecutive days. A total of 12 replicates of all the dilutions were tested. Probit analysis was carried out to empirically determine LOD in the clinical samples. The analytical specificity was determined by testing the suspensions of the target and non-target strains with concentrations between LOD and 10^5^ genome/mL in PBS.

### 2.4. Clinical Sampling and Processing

The study was conducted in accordance with the Declaration of Helsinki and approved by the research ethics committees of Istanbul University (13/08.06.2014) and Istanbul Medipol University (122/17.12.2013; 187/12.08.2014).

The clinical specimens were collected at IMU, IU and NKU Hospitals. No restrictions were placed on age, gender, medications or known pharmaceutical therapies. Between December 2015 and April 2018, 1929 patients in the intensive care unit (ICU) (63.6%) and non-ICU settings were enrolled in the study. Samples were obtained from patients with suspected bloodstream (532), central nervous system (216), gastrointestinal (190) and respiratory (991) infections.

CSF, stool, oropharyngeal and nasopharyngeal swabs, nasopharyngeal wash/aspirate, sputum, and bronchoalveolar lavage samples were used for both the culture and molecular analyses. Blood specimens for molecular analysis were collected in EDTA blood tubes simultaneously whenever blood cultures were taken. From the signal-positive blood culture tubes, 500 µL samples were also analyzed using the Bio-Speedy^®^ qPCR panel for the bloodstream infections.

Routine clinical microbiology protocols were applied as the gold standard for the detection of bacterial, fungal, and parasitic pathogens. FTD respiratory pathogen assays (Fast Track Diagnostics, Luxembourg), Allplex™ gastrointestinal and meningitis panel assays (Seegene, Seoul, Republic of Korea), and the qPCR protocols of the Centers for Disease Control and Prevention (CDC) and World Health Organization (WHO) were used as the gold standard for the detection of viral pathogens.

## 3. Results

No Bio-Speedy^®^ multiplex qPCR panels produced positive results for the samples spiked with the potential cross-reacting pathogens. LOD and repeatability of the assays were in the range of 10–100 pathogens/mL and 96–100%, respectively (Table 1, Table 2, Table 3, Table 4 and Table 5).

In the respiratory panel, a total of 243 true positives and 393 true negatives were recorded for Influenza A virus, with only 8 false positives and 8 false negatives, yielding a sensitivity of 97.3% and specificity of 96.3% (Table 1). Similarly, the GI panel achieved a sensitivity of 94.3% and specificity of 97.9%, effectively detecting common pathogens such as *Salmonella* spp., Norovirus, and *Clostridium difficile* (Table 2). The CNS panel demonstrated high sensitivity (96.4%) and specificity (96.8%) across targets including *Streptococcus pneumoniae*, *Herpes simplex virus*, and *Cryptococcus neoformans* (Table 3). The blood and blood culture panels showed improved performance, with specificity of >98% for all tested targets and sensitivity ranging from 82.0% in whole blood to 97.1% in blood culture-positive samples (Table 4 and Table 5).

The analysis of the true +/− and false +/− results is presented in Table 1, Table 2, Table 3, Table 4 and Table 5. The sensitivity, specificity, positive predictive value (PPV) and negative predictive value (NPV) of the qPCR panels are shown in Table 6. The statistics of the single and co-infections are given in Table 7. The qPCR panels detected all the agents of co-infections. No co-infection was detected for the CNS samples. The statistics of the detected off-panel organisms are illustrated in Table 8. There was no detected off-panel organism for the gastrointestinal panel.

We must emphasize that the summary statistics presented in the Abstract were calculated using the aggregated performance values from these tables, notably from the comparative analysis against culture and reference PCR methods. Furthermore, the qPCR panels were able to detect all reported co-infections (Table 7) and identified several off-panel organisms (Table 8), although the CNS panel did not detect any co-infections.

## 4. Discussion

The performance of the Bio-Speedy^®^ qPCR syndromic testing panels was sufficient and proved effective in terms of the identification of the causative pathogens tested.

In fact, the platforms used in the molecular panels evaluated in our study have related pros and cons seen in semi-automated platforms compared with fully automated ones [BioFire Diagnostics [6], Luminex (now named DiaSorin [7]), Seegene [8], Fast Track Diagnostics (now owned by Siemens Healthineers [9]), GenMark Diagnostics (now named cobas eplex system [10]), and Aus Diagnostics [11]] (Table 9). The semi-automated systems allow microbiology laboratories to manage numerous samples, as opposed to automated platforms, which are merely for emergency and point-of-care diagnostics. The total assay duration is between 1 and 2.5 h for the fully automated platforms, and between 2.5 and 4.5 h for the semi-automated platforms. While fully automated systems provide quick assessments, the semi-automated platforms provide more tests per run with a much lower cost per sample [12]. Thus, screening of a wider range of pathogens per assay is feasible.

All the off-panel targets of the Bio-Speedy^®^ are clinically relevant pathogens [13]. *Corynebacterium* spp., *C. lusitaniae* and *Micrococcus* spp. are only in GenMark’s sepsis panel. *S. maltophilia* is in the sepsis panels of GenMark and Seegene. *Acinetobacter* spp. is only in Luminex’s sepsis panel. *M. catarrhalis* is only in the respiratory panel of Fast Track Diagnostics. In contrast, *T. pallidum* and *Brucella* spp. are not in any of the CNS panels. When the prevalence and pathogen-specific treatment options of the off-panel microorganisms [13] were evaluated together, it was concluded that the exclusion of *Candida* spp., *Acinetobacter* spp., and *S. maltophilia* in the Bio-Speedy^®^ sepsis panel is a drawback.

In the assessed method in our study, the sensitivity and specificity for blood culture (BC), CNS, GI, and respiratory samples were better than previously reported (Table 9). In fact, Seegene’s Magicplex™ Sepsis Real-time Test (MSR) [8], Roche’s LightCycler^®^ SeptiFast Test (LSF) [14] and the Bio-Speedy^®^ qPCR sepsis panel are the only commercially available options for direct pathogen screening in whole blood via qPCR. Highly variable diagnostic performance has been reported for the most widely studied LSF [15]. There are no considerable differences between the specificities of the three platforms. The sensitivity of MSR [16] is much lower than that of the Bio-Speedy^®^ qPCR sepsis panel and LSF. Lengthy hands-on time (ranging from 5 to 7 h) impedes the LSF efficacy. Whole blood identification may complement BC, with better results in less than 3 h. Rapid detection of a causative pathogen of bacteremia and severe sepsis leads to immediate initiation of a proper antibiotic regimen, subsequently reducing complication rates and reducing healthcare costs [17,18,19,20].

The main limitations of our study are summarized in the following points: 1. The analysis was conducted using archived clinical samples collected in a single country, which may limit the generalizability of results to other geographic regions or healthcare settings. 2. Although the panels covered a broad range of pathogens, some clinically relevant microorganisms such as *Epstein Barr virus* and *parvovirus B19* were not included in the CNS panel, as well as *Candida* spp., *Acinetobacter* spp., and *S. maltophilia*. 3. While the panels demonstrated strong analytical performance, some off-panel organisms were not detected, reflecting the inherent limitations of fixed multiplex designs. 4. The diagnostic comparison was limited to standard culture and reference PCR methods; metagenomic sequencing was not used to resolve discordant results.

## 5. Conclusions

The ability to simultaneously detect possible pathogens that contribute to the constellation of symptoms that patients could suffer from makes syndromic testing a valuable method for use in clinical practice. The number and variety of samples evaluated in a single run using Bio-Speedy^®^ enable the rapid diagnosis of various and relevant infections. Such a method of testing, which aids in the timely diagnosis of infectious diseases, influences critical decisions regarding antimicrobial therapy, improves stewardship, and assists greatly in managing infections. Having said that, microorganisms such as *Candida* spp., *Acinetobacter* spp., and *S. maltophilia*, should be added to the panel to improve detection.

## Figures and Tables

**Table 1 diagnostics-15-01228-t001:** Multiple targets in 8-well qPCR strips of Bio-Speedy^®^ qPCR panel for respiratory tract samples, and results of the clinical performance study.

Multiplex Reactions		Respiratory Panel	LOD mL^−1^	Precision	True +	False +	True -	False -
1A	FAM	Influenza A virus	66	100%	243	8	393	8
	HEX	Internal Control	-					
	ROX	Influenza A H1 virus	83	100%	177			
	CY5	Influenza B virus	88	98%	113	6		5
1B	FAM	Coronavirus 229E	62	96%	12			
	HEX	Coronavirus OC43	68	100%	7			
	ROX	Coronavirus NL63	94	98%	9			
	CY5	Coronavirus HKU1	86	100%	2			
1C	FAM	Parainfluenza 1 virus	75	96%	8			
	HEX	Parainfluenza 2 virus	91	100%	10			
	ROX	Parainfluenza 3 virus	53	98%	1			
	CY5	Parainfluenza 4 virus	88	100%	6			
1D	FAM	Metapneumovirus	64	96%	19			
	HEX	MERS-CoV	92	100%	0			
	ROX	Respiratory syncytial virus A/B	97	100%	8			
	CY5	Rhinovirus	97	98%	18			
1E	FAM	Bocavirus	83	96%	17			
	HEX	Enterovirus	71	100%	22			1
	ROX	Parechovirus	59	100%	4			
	CY5	Adenovirus	62	98%	28	1		1
1F	FAM	*Legionella pneumophila*	54	100%	0			
	HEX							
	ROX	*Mycoplasma pneumoniae*	62	100%	9			
	CY5	*Chlamydophila pneumoniae*	58	98%	2			
1G	FAM	*Haemophilus influenzae*	48	96%	11			
	HEX							
	ROX							
	CY5	*Streptococcus pneumoniae*	43	98%	31			1
1H	FAM	*Bordetella pertussis*	47	98%	2			
	HEX							
	ROX	*Bordetella parapertussis*	63	96%	0			
	CY5	*Bordetella holmesii*	42	96%	0			

**Table 2 diagnostics-15-01228-t002:** Multiple targets in 8-well qPCR strips of Bio-Speedy^®^ qPCR panel for gastrointestinal samples, and results of the clinical performance study.

Multiplex Reactions		Gastrointestinal Panel	LOD mL^−1^	Precision	True +	False +	True -	False -
1A	FAM	Sapovirus (GI/GII/GIV/GV)	98	98%	1		95	
	HEX	Internal Control	-	-				
	ROX							
	CY5							
1B	FAM	*Yersinia enterocolitica*	74	98%				
	HEX	*Plesiomonas shigelloides*	62	100%	2			
	ROX	*Entamoeba histolytica*	58	100%				
	CY5	*Cryptosporidium* spp.	67	100%	1			
1C	FAM	*Giardia lamblia*	92	100%	2			
	HEX							
	ROX							
	CY5	*Cyclospora cayetanensis*	88	96%	1			
1D	FAM	Astrovirus	46	98%	2			
	HEX	Norovirus (GI/GII)	31	98%	6			
	ROX	Rotavirus (A)	24	98%	6			1
	CY5	Adenovirus	62	100%	3			
1E	FAM	*Salmonella* spp.	24	100%	14	1		1
	HEX	*Campylobacter* spp.	31	96%	10	1		1
	ROX	*Vibrio parahaemolyticus*	28	96%				
	CY5	*Vibrio cholerae*	63	96%				
1F	FAM	Enteroinvasive *E.coli*	66	100%	2			
	HEX							
	ROX	Enteroaggregative *E. coli*	54	98%	8			
	CY5	Shiga toxin producing *E.coli*	54	98%	2			
1G	FAM	Enteropathogenic *E.coli*	74	98%	27			2
	HEX							
	ROX							
	CY5	Enterotoxigenic *E.coli*	83	98%	2			
1H	FAM	*Clostridium difficile*	62	98%	11			1
	HEX	*C. difficile* toxin B	68	100%				
	ROX	*C. difficile* toxin A	52	98%	3			
	CY5	*C. difficile* binary toxin A/B	38	98%				

**Table 3 diagnostics-15-01228-t003:** Multiple targets in 8-well qPCR strips of Bio-Speedy^®^ qPCR panel for central nervous system samples, and results of the clinical performance study.

Multiplex Reactions		Central Nervous System Panel	LOD mL^−1^	Precision	True +	False +	True -	False -
1A	FAM	*Mycobacterium tuberculosis*	72	96%			182	
	HEX	Internal Control	-	-				
	ROX							
	CY5							
1B	FAM	*Listeria monocytogenes*	68	96%	2			
	HEX							
	ROX	*Neisseria meningitidis*	91	98%	2			
	CY5	*Streptococcus pneumoniae*	74	96%	13	6		
1C	FAM	*Haemophilus influenzae*	71	96%	1			
	HEX							
	ROX	*Streptococcus agalactiae*	59	98%	0			
	CY5	*Escherichia coli* K1	77	96%	1			
1D	FAM	Cytomegalovirus	63	96%	1			
	HEX	Enterovirus	81	96%	3			1
	ROX	Parechovirus	82	100%	1			
	CY5	Varicella Zoster Virus	79	98%	1			
1E	FAM	Human Herpesvirus 6	91	96%	1			
	HEX							
	ROX	Human Herpesvirus 7	66	96%	0			
	CY5	Human Herpesvirus 8	56	98%	0			
1F	FAM	Herpes simplex virus 1	32	100%	1			
	HEX							
	ROX							
	CY5	Herpes simplex virus 2	24	98%	0			
1G	FAM	*Cryptococcus gattii*	33	98%	0			
	HEX							
	ROX							
	CY5	*Cryptococcus neoformans*	57	98%	0			

**Table 4 diagnostics-15-01228-t004:** Multiple targets in 8-well qPCR strips of Bio-Speedy^®^ qPCR panel for bloodstream samples, and results of the clinical performance study.

Multiplex Reactions		Sepsis Panel	LOD mL^−1^	Precision	True +	False +	True -	False -
1A	FAM	*Staphylococcus* spp.	92	100%	39	5	397	25
	HEX	Internal control	-	-	-			
	ROX	*Brucella* spp.	86	96%	1			
	CY5	*Listeria monocytogenes*	88	96%	1			
1B	FAM	*Staphylococcus aureus*	94	96%	4			
	HEX	*Candida albicans*	98	96%	1			
	ROX	vanA/vanB—Vancomycin resistance	52/68	96%	5			
	CY5	*Candida krusei*	68	98%	2			
1C	FAM	*Pseudomonas aeruginosa*	42	100%	4			
	HEX	*Aspergillus*/*Fusarium*/*Trichosporon* spp.	86/88/94	96%				
	ROX	*Klebsiella pneumoniae*	82	96%	12	1		
	CY5	*Acinetobacter baumannii*	96	96%	6			
1D	FAM	*Haemophilus influenzae*	64	96%	1			
	HEX	*Klebsiella oxytoca*	94	96%	3			
	ROX	*Candida parapsilosis*	68	98%	0			
	CY5	OXA-48—Carbapenem resistance	74	100%	3			
1E	FAM	KPC—Carbapenem resistance	78	98%	1			
	HEX	NDM—Carbapenem resistance	82	96%	5			
	ROX	VIM—Carbapenem resistance	78	98%	1			
	CY5	IMP—Carbapenem resistance	92	96%	1			
1F	FAM	mcr-1—Colistin resistance	88	96%				
	HEX	*Candida glabrata*	86	96%	1			
	ROX	mecA/mecC—Methicillin resistance	92/97	98%	1			
	CY5	*Candida tropicalis*	76	96%	4			
1G	FAM	*Enterococcus* spp.	78	98%	8			
	HEX	*Pseudomonas* spp.	84	96%	4			
	ROX	*Enterobacteriaceae*	88	96%	29			
	CY5	*Streptococcus* spp.	92	98%	4			
1H	FAM	OXA-23/51/58—Carbapenem resistance	78/62/66	98%	4			
	HEX	*Escherichia coli*	68	100%	9	1		
	ROX	*Neisseria meningitidis*	86	96%	0			
	CY5	*Streptococcus pneumoniae*	76	96%	2			

**Table 5 diagnostics-15-01228-t005:** Multiple targets in 8-well qPCR strips of Bio-Speedy^®^ qPCR panel for positive blood culture samples, and results of the clinical performance study.

Multiplex Reactions		Sepsis Panel	LOD mL^−1^	Precision	True +	False +	True -	False -
1A	FAM	*Staphylococcus* spp.	6	100%	60	3	401	4
	HEX	Internal control	-	-	-			
	ROX	*Brucella* spp.	8	100%	1			
	CY5	*Listeria monocytogenes*	6	100%	1			
1B	FAM	*Staphylococcus aureus*	7	100%	4			
	HEX	*Candida albicans*	6	100%	1			
	ROX	vanA/vanB—Vancomycin resistance	6	100%	5			
	CY5	*Candida krusei*	7	100%	2			
1C	FAM	*Pseudomonas aeruginosa*	4	100%	4			
	HEX	*Aspergillus*/*Fusarium*/*Trichosporon* spp.	8/8/2009	100%				
	ROX	*Klebsiella pneumoniae*	8	100%	12			
	CY5	*Acinetobacter baumannii*	8	100%	6			
1D	FAM	*Haemophilus influenzae*	4	100%	1			
	HEX	*Klebsiella oxytoca*	8	100%	3			
	ROX	*Candida parapsilosis*	9	100%	0			
	CY5	OXA-48—Carbapenem resistance	6	100%	3			
1E	FAM	KPC—Carbapenem resistance	6	100%	1			
	HEX	NDM—Carbapenem resistance	5	100%	5			
	ROX	VIM—Carbapenem resistance	6	100%	1			
	CY5	IMP—Carbapenem resistance	7	100%	1			
1F	FAM	mcr-1—Colistin resistance	8	100%				
	HEX	*Candida glabrata*	5	100%	1			
	ROX	mecA/mecC—Methicillin resistance	6/8	100%	1			
	CY5	*Candida tropicalis*	7	100%	4			
1G	FAM	*Enterococcus* spp.	9	100%	8			
	HEX	*Pseudomonas* spp.	6	100%	4			
	ROX	*Enterobacteriaceae*	5	100%	29			
	CY5	*Streptococcus* spp.	5	100%	4			
1H	FAM	OXA-23/51/58—Carbapenem resistance	8/9/2005	100%	4			
	HEX	*Escherichia coli*	7	100%	9			
	ROX	*Neisseria meningitidis*	6	100%	0			
	CY5	*Streptococcus pneumoniae*	6	100%	2			

**Table 6 diagnostics-15-01228-t006:** Sensitivity, specificity, positive predictive value (PPV) and negative predictive value (NPV) of the Bio-Speedy^®^ qPCR panels.

Panel	Sensitivity	Specificity	PPV	NPV
Respiratory	97.3%	96.3%	97.5%	96.1%
Gastrointestinal	94.3%	97.9%	98.0%	94.1%
Central Nervous System	96.4%	96.8%	81.8%	99.5%
Blood	82.0%	98.3%	94.2%	94.1%
Blood culture	97.1%	99.3%	97.8%	99.0%

**Table 7 diagnostics-15-01228-t007:** Statistics of the single and co-infections.

Panel	Co-Infection			Single-Infection	
	Agent	True +		True +	False -
Respiratory	*S. pneumoniae* + RSV	6	15	567	16
	*S. pneumoniae* + Rhinovirus	3			
	RSV + Parainfluenza 1	1			
	RSV + Adenovirus	1			
	RSV + Rhinovirus	1			
	Parainfluenza 1 + Bocavirus	1			
	Bocavirus + Rhinovirus	1			
	RSV + Parainfluenza 1 + Rhinovirus	1			
Gastrointestinal	*Campylobacter* spp. + Adenovirus	2	13	87	6
	*Salmonella* spp. + Astrovirus	2			
	Rotavirus + Enteroaggregative *E. coli*	2			
	*P. shigelloides* + Norovirus (GI/GII)	2			
	Norovirus (GI/GII) + Enteroaggregative *E. coli*	2			
	*Salmonella* spp. + Adenovirus	1			
	*Campylobacter* spp. + Rotavirus	1			
	Enteroaggregative *E. coli* + Astrovirus	1			
Blood	*A. baumannii* + *P. aeruginosa*	1	11	103	25
	*A. baumannii* + *P. aeruginosa* + *Staphylococcus* spp.	1			
	*Enterococcus* spp. + *Staphylococcus* spp.	1			
	*Enterococcus* spp. + *K. pneumoniae*	3			
	*E. coli* + *K. pneumoniae*	1			
	*Enterobacteriaceae* + *K. pneumoniae*	4			
Blood culture	Same as the blood agents	11	11	124	4

**Table 8 diagnostics-15-01228-t008:** Statistics of the detected off-panel organisms.

Panel	Target	Positives	Off-Panel Agent % in the Culture Positives
Blood	*Stenotrophomonas maltophilia*	2	4.3%
	*Micrococcus* spp.	1	
	*Corynebacterium striatum*	1	
	*Candida lusitaniae*	1	
	*Acinetobacter lwoffii*	1	
Respiratory	*Moraxella catarrhalis*	7	1.2%
Central nervous system	*Treponema pallidum*	1	7.1%
	*Brucella* spp.	1	

**Table 9 diagnostics-15-01228-t009:** Bacterial/fungal/viral/parasitic (Bac/Fun/Vir/Par) targets and the other testing properties of commercial syndromic testing panels for respiratory (RP), gastrointestinal (GI), central nervous system (CNS), and blood and blood culture (BC) samples.

Brand	Panel	Sensitivity/Specificity	Targets	Duration	Run Capacity per Instrument	
Biofire	RP	97.1%/99.3%	3Bac/17Vir	1–1.5 h	Full automation	1 sample, 1 type of panel
	GI	98.5%/99.2%	13Bac/5Vir/4Par			
	CNS	94.2%/99.8%	6Bac/7Vir/3Fun			
	BC	98%/99.9%	19Bac/5Fun/3Res			
Luminex	RP	95.2%/99.6%	2Bac/17Vir	3.54 h	Semi-automation	24 samples, 1 type of panel
	GI	94.3%/98.5%	9Bac/3Vir/3Par			
	BC	89.6–90.5%/98.9–100%	22Bac/9Res	2–2.5 h	Full automation	1 sample, 1 type of panel
Seegene	RP	82.8–100%/95.5–100%	7Bac/19Vir	2.5–3.5 h	Semi-automation	8–10 samples, all types of panels
	GI	93.3–100%/99.2–100%	14Bac/6Vir/5Par			
	CNS	100%/100%	6Bac/12Vir			
	Blood	29%/95%	24Bac/6Fun/3Res			
Fast	RP	>98%	19Vir/12Bac	2.5–3.5 h	Semi-automation	8–10 samples, all types of panels
	GI	>99.5%	6Vir/6Bac/3Par			
	CNS	>97.8%	6Vir/3Bac			
GenMark	RP	>97.4%	15Vir/2Bac	2 h	Full automation	3 samples, 1 type of panel
	BC	>94%	41Bac/14Res/15Fun			
AUS	RP	>93.5%/99.7%	9Vir/5Bac	4.5 h	Semi-automation	24 samples, all types of panels
	GI	>94.7%/98.9%	6Bac/5Vir/3Par			

## Data Availability

Data regarding molecular testing are presented in the tables of the paper. Additional data are available from the authors. Due to privacy and ethical restrictions other data are not publicly available.

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
