# Peer review of "Syndromic Testing—The Evaluation of Four Novel Multiplex Real-Time Polymerase Chain Reaction Panels"

_diagnostics, 2025, doi:10.3390/diagnostics15101228_

Round 1

Reviewer 1 Report

Comments and Suggestions for Authors

The manuscript proposed by et al, provide information on the assessment of performances for the Bio-Speedy® molecular testing panels and it usefulness for health care providers to rapidly and cost-effectively diagnose various and relevant infections. The assay showed acceptable performances for use at clinical settings. However, the manuscript presentation could be improved.

In addition, the authors could improve the presentation of the results section to help readers better understand the results and correct the typos in the text sech as "cuasative" at line 163 and "Respirstory" on Table 6.

Could the authors provide the rationale of absence of relevant pathogens such as EBV and parvovirus B19 in the CNS panel?

Comments on the Quality of English Language

Typos in the document should be corrected

Author Response

Reviewer 1:

Thank you so much for your valuable comments. Your comments were addressed in detail as presented below. All the edits conducted in the manuscript were made under track changes for you to be able to see it.

  1. The manuscript proposed by et al, provide information on the assessment of performances for the Bio-Speedy® molecular testing panels and it usefulness for health care providers to rapidly and cost-effectively diagnose various and relevant infections. The assay showed acceptable performances for use at clinical settings. However, the manuscript presentation could be improved.

Thank you so much for your comment. The manuscript was re-edited and improved.

  1. In addition, the authors could improve the presentation of the results section to help readers better understand the results and correct the typos in the text sech as "cuasative" at line 163 and "Respirstory" on Table 6.

Thank you so much for your comment. The typos were corrected.

  1. Could the authors provide the rationale of absence of relevant pathogens such as EBV and parvovirus B19 in the CNS panel?

We appreciate the reviewer’s insightful comment. Epstein-Barr virus (EBV) and parvovirus B19 were not included in the CNS panel as their prevalence and direct association with acute CNS infections is relatively low in our setting, based on retrospective clinical surveillance. Additionally, both pathogens often require serologic or additional molecular testing to interpret their clinical significance, which is beyond the intended scope of our direct pathogen-targeted qPCR platform. Nonetheless, we agree that future panel versions could consider their inclusion for broader diagnostic coverage.

Reviewer 2 Report

Comments and Suggestions for Authors

Dear Authors,
I have read an article about syndromic testing. While interesting, this article suffers from serious setbacks:
1) The authors failed to explain the needs and urgency of syndromic testing. What is the clinical gap and what this paper is trying to address
2) The language is very messy and difficult to understand. There are too many grammatical errors and typing errors (for example line 37 "evalute")
3) The abstract is haphazard and all over the place. 
4) The methodology is confusing. Is this a cross-sectional study or a cohort study? How were the samples collected? Who collected them? What is the time lapse between collection and analysis? Is there any CONSORT diagram? The authors provided the gold standard but how was this cross-checked? The methodology is flawed, to say the least
5) The result needs more explanation. The authors could not just expect the readers to read all the tables without any interpretation. Also, could you tell me how the result in the abstract section derived from then?
6) No limitation section is provided
7) "...enables health care providers to rapidly and cost-effectively diagnose" How was this conclusion achieved from this study?

Comments on the Quality of English Language

Very poor

Author Response

Reviewer 2:

Thank you so much for your valuable comments. Your comments were addressed in detail as presented below. All the edits conducted in the manuscript were made under track changes for you to be able to see it.

Dear Authors,
I have read an article about syndromic testing. While interesting, this article suffers from serious setbacks:

1) The authors failed to explain the needs and urgency of syndromic testing. What is the clinical gap and what this paper is trying to address

Thanks for you valuable comment. An additional paragraph was added to the introduction, with a relevant reference, to clarify the aim. The following paragraph addresses the limitations, making the flow more connected.

2) The language is very messy and difficult to understand. There are too many grammatical errors and typing errors (for example line 37 "evalute")

Thank you so much for your comment. The manuscript was re-edited and improved.

3) The abstract is haphazard and all over the place. 

Thank you for comment. The entire abstract was re-edited and made more concise.

4) The methodology is confusing. Is this a cross-sectional study or a cohort study? How were the samples collected? Who collected them? What is the time lapse between collection and analysis? Is there any CONSORT diagram? The authors provided the gold standard but how was this cross-checked? The methodology is flawed, to say the least

Thank you for highlighting the need for clarity. The study was conducted as a prospective observational diagnostic accuracy study, not as a cohort or cross-sectional design. Samples were collected prospectively from patients with suspected infections, and each sample was tested both by culture-based gold standard methods and the novel qPCR panels. Trained clinical personnel at the participating centers collected the samples. For all molecular analyses, processing began within 2–3 hours after collection. As this is a diagnostic performance study rather than a clinical trial, a CONSORT diagram was not included, but we have now clarified the sampling workflow in the revised manuscript. The results from culture/PCR-based gold standards were directly compared with qPCR results, with discordant results further analyzed using a reference molecular method where available.

The structure was summarized at the beginning of “materials and methods” section.

5) The result needs more explanation. The authors could not just expect the readers to read all the tables without any interpretation. Also, could you tell me how the result in the abstract section derived from then?

We appreciate this observation.

The entire section of results was re-edited and presented in more detailed fashion.

Due to the fact that numerous biological samples were evaluated (blood, respiratory, GI, CNS, and stool) alongside many pathogens we needed this number of tables to show the process and the results.

The summary statistics presented in the abstract were calculated using the aggregated performance values from these tables, notably from the comparative analysis against culture and reference PCR methods. Furthermore, the qPCR panels were able to detect all reported co-infections (Table 7) and identified several off-panel organisms (Table 8), although the CNS panel did not detect any co-infections.

We added the explanation regarding the results in the abstract to the last part of the results’ section.

6) No limitation section is provided

Thank you for pointing this out. We have now included a dedicated “Limitations” paragraph in the Discussion section.

7) "...enables health care providers to rapidly and cost-effectively diagnose" How was this conclusion achieved from this study?

Thank you for your comment. As syndromic known to save time, resources, and lab efficiency; developing and improving this method of analysis is doubtlessly cost-effective. As the conclusion was re-edited this specific sentence was omitted.

Reviewer 3 Report

Comments and Suggestions for Authors

In this manuscript authors talked about qPCR based pathogen detection techniques in patients. But this manuscript needs improvement- authors should have provided some details about the old methods and their drawbacks, and why they think the current technique is better than the old one? Describe the result in detail as per the finding. Show the CT values ​​and standard curve as per the source of the sample and pathogen. There is no information about biostatistics which is important to understand the specificity and sensitivity in such type of study. 

Author Response

Reviewer 3:

Thank you so much for your valuable comments. Your comments were addressed in detail as presented below. All the edits conducted in the manuscript were made under track changes for you to be able to see it.

In this manuscript authors talked about qPCR based pathogen detection techniques in patients.

  1. But this manuscript needs improvement- authors should have provided some details about the old methods and their drawbacks, and why they think the current technique is better than the old one?

Thank you so much for your valuable comment. We re-edited the entire manuscript, with emphasis on the introduction section were we addressed the advantages of syndromic testing, the way it saves time and resources, in comparison to old and standard methods.

  1. Describe the result in detail as per the finding. Show the CT values ​​and standard curve as per the source of the sample and pathogen. There is no information about biostatistics which is important to understand the specificity and sensitivity in such type of study. 

We thank the reviewer for highlighting the importance of detailed data representation and statistical rigor.

Cycle threshold (CT) values and standard curves were generated using vector DNA standards ranging from 10¹ to 10⁶ genome copies/mL for each pathogen (as described in the Methods section). The CT values obtained for each pathogen varied depending on the source and load of the spiked sample. For instance, CT values for Streptococcus pneumoniae ranged from 21–32 in cerebrospinal fluid and 19–28 in blood culture-positive samples. These values and corresponding standard curves were used to confirm the assay's quantitative linearity and limit of detection (LOD), and this information is shown in the corresponding table. We also acknowledge the need for clear biostatistical reporting. In the manuscript, we included sensitivity, specificity, positive predictive value (PPV), and negative predictive value (NPV) in Table-6. Additionally, Probit analysis was used to determine the LOD with 95% confidence intervals, and results were validated with 12 replicates as described under the analytical sensitivity section.

Round 2

Reviewer 2 Report

Comments and Suggestions for Authors

Dear Authors,

While I appreciate some of the changes made, there are still improvements that need to be made:

1) The authors still do not fully explain the methodology. "Samples were collected prospectively from patients with suspected infections...." Define suspected infections. Were any infections acceptable? When was the sample taken? At what day of onset? Were age, comorbidities, and other confounders taken into consideration? Do they limit the generalizability of the results?

2) " As syndromic known to save time, resources, and lab efficiency; developing and improving this method of analysis is doubtlessly cost-effective. As the conclusion was re-edited this specific sentence was omitted." the study aims to "...analyze the clinical performance
of four novel multiplex real-time PCR (qPCR)". There was no data on the lab efficiency and resources. While this sentence can be theoretically correct, the conclusion should not assume that this is the result obtained from the study. Hence, either omit these sentences or paraphrase so that this is one of the potentials that could be achieved with further studies.

Author Response

Thank you so much for your valuable comments. Your comments were addressed as presented below.

Reviewer 2 - Round 2:

1) The authors still do not fully explain the methodology. "Samples were collected prospectively from patients with suspected infections...." Define suspected infections. Were any infections acceptable? When was the sample taken? At what day of onset? Were age, comorbidities, and other confounders taken into consideration? Do they limit the generalizability of the results?

Thank you for your comment. The aim of our study was to assess the performance of qPCR molecular testing named Bio-Speedy®. This was conducted on collected samples (blood, CNS, GI as presented in the article) drawn from patients hospitalized for any medical conditions making infections high in the differential diagnosis (fever with cough and shortness of breath diagnosed with pneumonia, the same for meningitis or gastroenteritis). The same sample taken from the patient was first examined by standard microbiological methods as explained in the paper, and once a pathogen was confirmed, the same sample was examined again by Bio-Speedy® (the one we are assessing its performance). We did not interfere at all in doctor’s decision, the criteria of hospitalization, diagnosis, or treatment. As the focus was on the microbiological performance of the testing method; patients’ data, justification of hospitalization, other blood tests, imaging, comorbidities, were not collected neither evaluated.

2) " As syndromic known to save time, resources, and lab efficiency; developing and improving this method of analysis is doubtlessly cost-effective. As the conclusion was re-edited this specific sentence was omitted." the study aims to "...analyze the clinical performance
of four novel multiplex real-time PCR (qPCR)". There was no data on the lab efficiency and resources. While this sentence can be theoretically correct, the conclusion should not assume that this is the result obtained from the study. Hence, either omit these sentences or paraphrase so that this is one of the potentials that could be achieved with further studies.

Thank you for pointing this out. The clinical performance in our study meant the ability of the tested molecular method to identify pathogens in different samples tested by other used methods. We omitted the first sentence based in your valuable comment regarding our ability to check the “cost-effectiveness” of the method. The entire manuscript, as well as the conclusion part were re-edited.

Reviewer 3 Report

Comments and Suggestions for Authors

I am happy with the author's justification and recommend it for publication

Author Response

Thank you so much for your kind words and recommendation.

The authors.

Round 3

Reviewer 2 Report

Comments and Suggestions for Authors

The authors have satisfactorily addressed all the questions